# Dynamic Multi-Attention Dehazing Network with Adaptive Feature Fusion

Donghui Zhao [1,*], Bo Mo [1], Xiang Zhu [2], Jie Zhao [1,3], Heng Zhang [4], Yimeng Tao [1] and Chunbo Zhao [1]

[1] Beijing Institute of Technology, Beijing 100081, China
[2] Beijing Building Materials Research Institute Co., Ltd., Beijing 100041, China
[3] North Navigation Control Technology Co., Ltd., Beijing 100176, China
[4] Shanghai Electro-Mechanical Engineering Institute, Shanghai 201109, China
[*] Correspondence: zhaodonghui99@foxmail.com

**Abstract:** This paper proposes a Dynamic Multi-Attention Dehazing Network (DMADN) for single image dehazing. The proposed network consists of two key components, the Dynamic Feature Attention (DFA) module, and the Adaptive Feature Fusion (AFF) module. The DFA module provides pixel-wise weights and channel-wise weights for input features, considering that the haze distribution is always uneven in a degenerated image and the value in each channel is different. We propose an AFF module based on the adaptive mixup operation to restore the missing spatial information from high-resolution layers. Most previous works have concentrated on increasing the scale of the model to improve dehazing performance, which makes it difficult to apply in edge devices. We introduce contrastive learning in our training processing, which leverages both positive and negative samples to optimize our network. The contrastive learning strategy could effectively improve the quality of output while not increasing the model's complexity and inference time in the testing phase. Extensive experimental results on the synthetic and real-world hazy images demonstrate that DMADN achieves state-of-the-art dehazing performance with a competitive number of parameters.

**Keywords:** dehazing; CNN; feature attention; feature fusion; contrastive learning

## 1. Introduction

Haze is a common atmospheric phenomenon caused by floating particles in the air. Due to the turbid medium, light propagation is hindered, and images taken in the haze are often subject to some degree of degradation. Input images captured in the hazy environment will affect the performance of dependable high-level computer vision systems (such as object detection [1,2] and scene understanding [3,4]). However, a dependable high-level computer vision system must work well with various kinds of interference [5,6]. It is a significant step for developing dehazing techniques to improve the robustness of high-level computer vision systems.

Previous works [7,8] has proposed the atmosphere scattering model to explain the process of hazy image generation. Specifically, it assumes that:

$$I(x) = J(x)t(x) + A(1 - t(x)) \tag{1}$$

where $I(x)$ and $J(x)$ are the degenerated hazy and clear images, $A$ is the atmosphere light intensity, and $t(x)$ is the medium transmission map. We also have $t(x) = e^{-\beta d(x)}$, where $\beta$ and $d(x)$ are the atmosphere scattering parameter and the scene depth, respectively.

Early dehazing methods [9–19] are based on priors in nature scenes; He et al. [12] proposed the dark channel prior (DCP) which is the masterpiece of the prior-based method. However, prior-based dehazing methods are not efficient in specific scenarios. In recent years, the Convolutional Neural Network (CNN) has been proven effective in dehazing [20–27]. DehazeNet [21] first reconstructs the haze-free image by estimating $A$

and $t(x)$ in the atmosphere scattering model. Because of the airlight albedo ambiguity [28], it is not easy to estimate the atmosphere light intensity and the medium transmission map. Moreover, it may generate artifacts in the images due to the cumulative error between the estimated and actual parameters, and it is also an expensive task to obtain the ground-truth medium transmission map and atmosphere light intensity. Unlike DehazeNet, some recent networks [25–27,29] try to directly find the mapping without resorting to the physical model and have achieved good performance. Nevertheless, most end-to-end methods ignore that haze is unevenly distributed in a degenerated image and treat pixel-wise features equally. In addition, the work [12] finds that for a hazy image $I(x)$, there is always a very low value in one channel. If we treat pixel-wise and channel-wise features equally, the information cannot be extracted efficiently. Considering the above problem, we propose the Dynamic Feature Attention (DFA) module, which includes a Channel Attention (CA) mechanism and a Pixel Attention (PA) mechanism. In this way, we can pay more attention to the important regions, which is helpful in expanding the representational ability of the network. Most of the latest works [24–27,29,30] only adopt convolutional layers with fixed kernels to build networks and it may generate over-smoothing artifacts and corrupted image textures in the dehazed image. To extract the input image's structural information better, we introduce deformable convolution [31] in our dehazing network. The experiment results have proven that the above measures significantly improve the dehazing performance.

The essence of the dehazing method is the image restoration task; we notice that most recent end-to-end dehazing networks are typically designed under an encoder–decoder framework [25,26,29,31–34]. Due to the downsampling operations, the spatial information from the high-resolution layer degenerates. To deal with this problem, previous works [31,35] first resize the non-adjacent features to the same dimension and fuse them via the bottleneck layer. However, they ignore that the density of information from shallow and in-depth features is different. Previous works [36,37] concatenate features from non-adjacent layers by using the strided convolutional layer. Although this method can fuse features with different scales, the scheme is ineffective for extracting information from input images. Recent works [32,38,39] proposed grid architectures to fuse features of adjacent layers, but this scheme cannot connect non-adjacent layers and cannot easily be applied to other networks. To remedy the missing spatial information, we propose an Adaptive Feature Fusion (AFF) module to exploit non-adjacent features based on the adaptive mixup operation [40].

Different from object detection CNNs, dehazing networks are notoriously computationally intensive. Most previous works concentrate on increasing the model scale to improve performance, making applying dehazing methods in edge devices challenging. For example, MSBDN [29] has 31 million parameters, and TDN [41] has 46.18 million parameters. On the other hand, most previous works only use clear images to optimize performance but ignore the negative samples. Recently, additional information from negative samples has been adopted in the training processing of some high-level computer vision [42–44]. It differs from only using reconstruction loss to approximate the ground-truth; they are two forces in contrastive learning. One force pulls the output close to the original haze-free images; the other pushes the result away from the degenerated images. This way, the contrastive learning strategy could improve output quality while adding no additional complexity and inference time in the testing phase.

The contributions of this paper are summarized as follows:

We propose a Dynamic Multi-Attention Dehazing Network for single image dehazing with competitive model complexity. Experiments have shown that our DMADN achieves state-of-the-art performance in terms of vision and can improve the robustness of high-level computer vision tasks.

We demonstrate that the extra pixel-wise and channel-wise weights provided by the Dynamic Feature Attention module are helpful for extracting features. Moreover, the Adaptive Feature Fusion module can effectively restore missing spatial information and help improve the performance of dehazing networks.

We adopt contrast learning in the training processing of DMADN, which leverages both the information from positive and negative samples. Experiments demonstrate that the contrast learning mechanism can significantly improve the dehazing performance without increasing the inference time.

## 2. Related Work

The imaging dehazing method is designed to resume the degraded images affected by haze, and it contains two categories: prior-based methods and learning-based methods.

**Prior-based Methods.** The handcraft prior-based dehazing methods [9–19] usually first estimate the $t(x)$ and $A$ in Equation (1) by leveraging the statistical characteristic of natural scenes. Tan [9] proposed an adaptive contrast dehazing algorithm by enhancing the local contrast of degraded images through the Markov Random Field. This method can remove most haze, but there are still some haloes and oversaturated areas in the results. Fattal [10] proposed an atmosphere light intensity estimation method that can restore sharpness and contrast. The work [11] estimates that the medium transmission map depends on the color-line prior in the regularity of the natural image. He et al. [12] find that there is always one channel with a low value in hazy natural scenes, termed dark channel prior (DCP). In this way, a relatively accurate medium transmission map can be found. Zhu et al. [13] proposed a dehazing method by estimating the scene depth through a color attenuation prior. Berman et al. [14] proposed a single dehazing method based on the non-local prior, and they approximate the original colors of a clear image through a few hundred distinct colors. One study [18] proposed an adaptive bi-channel priors method for image dehazing, which can rectify the incorrect estimations on atmosphere light intensity and the medium transmission map. To obtain more natural images, [19] proposed a kind of modified atmospheric scattering model. Although previous works have achieved promising results, prior-based methods have limitations in some particular environments (for example, DCP cannot handle the white region well). Furthermore, the assumptions of these priors are only satisfied in some scenes.

**Learning-based Methods**. With the powerful ability of CNN, most recent dehazing methods are based on "date drive". The learning-based dehazing method contains two categories: the methods based on transmission map estimation and end-to-end methods. The former first learns a non-linear mapping between the degenerated scene and its corresponding medium transmission map and then obtains the haze-free image via the physical model. Ren et al. [20] proposed a multi-scale dehazing network to estimate $t(x)$ in a coarse-to-fine manner. DehazeNet [21] reconstructed the clear image by estimating the atmosphere light intensity and the medium transmission map. Li et al. [22] proposed a residual-based network to predict the key parameters of the physical model. DCPDN [23] integrated the atmospheric scattering model with the CNN to jointly remove the haze. Another work [45] estimated the transmission map of the atmospheric scattering model by training an adversarial network model. However, the above dehazing methods may generate artifacts in the images due to the cumulative error between the estimated and 60ror, recent works [24–27,29,30,46–50] have tried to find the mapping between the input image and the ground-truth forthrightly. AOD-Net [24] developed a lightweight end-to-end convolutional neural network to remove haze. It has been applied to video dehazing and object detection tasks. In [25], a dehazing network was proposed using conditional generative adversarial learning. To improve the realism, it also employs the L1 regularized gradient prior. In [26], a multi-scale feature fusion-based end-to-end dehazing network called GFN was proposed. To utilize more information, GFN processes the hazy image with some image enhancement filters before feeding it into the network. To obtain more natural results, [27] introduced the Perceptual Index (PI) as a metric to evaluate the dehazing quality from the perceptual perspective. It also designed an enhanced pix2pix network and approached the dehazing process as an image translation task. To improve the dehazing performance, [30] introduced multi-scale patches in the dehazing process. In [41], the Detail Refinement sub-Net frequency was proposed, which is helpful to refine the high frequency

details in the dehazing process. Although previous end-to-end dehazing networks achieve good results in dehazing, they did not take advantage of the characteristic of haziness in their designing of networks. They also ignore the complexity of the models, which limits their practical application. Unlike these methods, we design a novel dehazing network that achieves a balance between weight and performance.

## 3. Materials and Methods

### 3.1. Dynamic Multi-Attention Dehazing Network

As shown in Figure 1, our DMADN includes three components, encoder module $G_E$, feature restoration module $G_R$, and decoder module $G_D$.

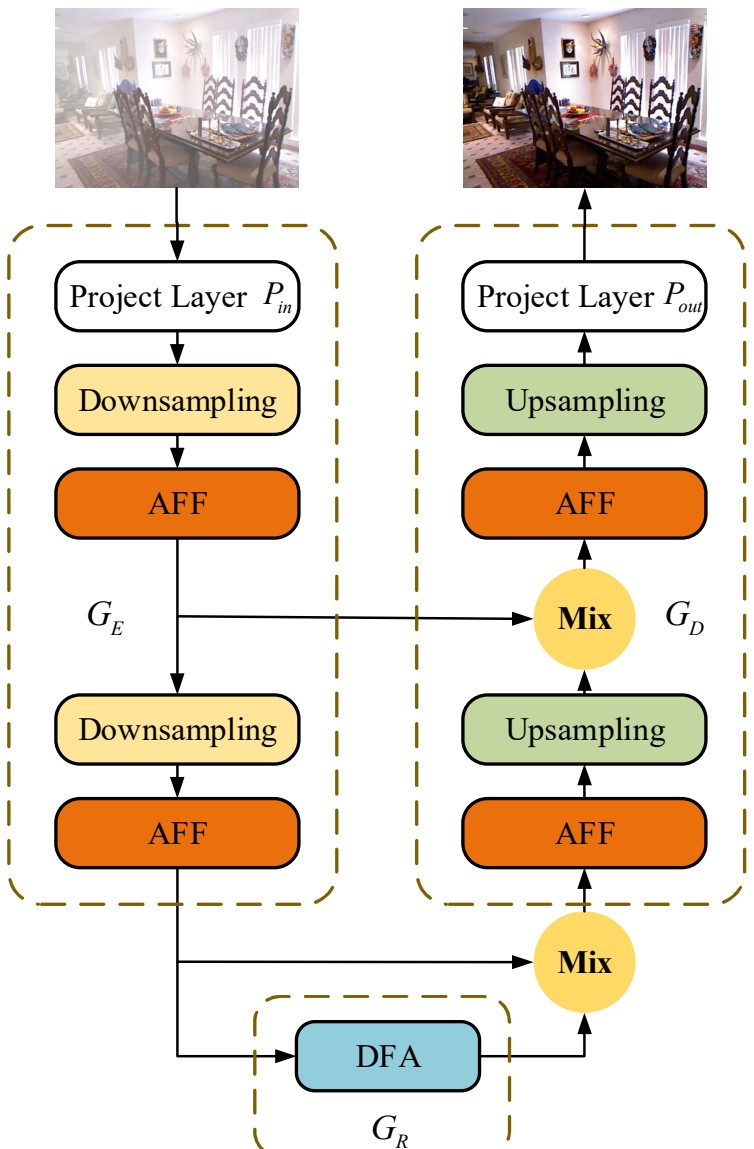

**Figure 1.** The structure of the proposed DMADN.

In DMADN, we first use two convolutional layers with strides 1 to extract the information of the hazy image $I \in R^{3 \times H \times W}$ and expand the channel to $F_0 \in R^{64 \times H \times W}$. To achieve a balance between performance and parameters of the network, we reduce the number of convolutional layers in encoder and decoder modules. In the encoder module, we only stack two convolutional layers with kernels $3 \times 3$ and strides 2 as downsampling layers to expand the channel capacity of $F_0$. In order to remedy the missing spatial information, we

place one AFF module after each downsampling layer. To achieve a balance between the number of parameters and the network's performance, we did not simply adopt multiple independent CNN models. Following works [29,51], we define only one DFA module with individual learnable parameters in the feature restoration module. To better extract information pertaining to semantic features, we let the input features pass through this defined DFA module several times. Through this strategy, the depth of the network can be increased without increasing the number of parameters, and the network's transformation capability is enhanced. To obtain the balance between inference time and performance, after experimental testing, we set the times value to 6. This approach of using shared parameters in the network slightly decreases performance compared to using separate parameter modules. However, its semantic extraction power is significantly enhanced compared to using the corresponding convolutional layer only once. As with the encoder module, in the decoder, we use two transposed convolutional layers with kernels $3 \times 3$ and strides 2 to restore spatial resolution and employ a convolutional layer with kernel $7 \times 7$ and stride 1 to obtain a haze-free image.

Inspired by [40], we designed a kind of adaptive mixup skip connection with learnable parameters to deal with the problem that information from the low-level is gradually degrading. Through this adaptive mixup skip connection, we can fuse the features from the encoder and decoder. The adaptive skip connection can be formulated as

$$\widetilde{F}^n = \sigma(\partial^n) * F_e^n + (1 - \sigma(\partial^n)) * F_d^n \tag{2}$$

where $F_e^n$, $F_d^n$ are features from the $n$th downsampling layer, $\widetilde{F}^n$ is the boosted feature, and $\partial^n$ is the balancing parameter. Differently from [40], we set the $\partial^n$ to be a learnable parameter, which can be optimized in the training process, and we use the Sigmoid function $\sigma$ to constrain the weight. As shown in Figure 1, we adopt the two above adaptive mixup skip connections, which have the learnable balancing parameter in our DMADN. Compared with simple addition or concatenation, our adaptive mixup skip connection supplies additional flexibility on fusing while adding ignorable parameters.

### 3.2. Dynamic Feature Attention Module

Most previous CNN-based dehazing methods ignore that haze is unevenly distributed on a degenerated image and treat pixel-wise features equally. In addition, [12] finds that for a hazy image, it is usually one channel that has a very low value. Inspired by [47,51], we propose the Dynamic Feature Attention (DFA) module. The DFA module provides pixel-wise weights, and channel-wise weights for input features simultaneously, which can allow the network to pay more attention to the regions with more important information.

As shown in Figure 2, the proposed DFA module includes three components: a Feature Extraction (FE) module, a Dynamic Feature Enhancement (DFE) module, and a Feature Attention Mechanism (FAM) module. The DFA module is based on residual block structure, which can help DMADN to ignore less important features and improve training stabilization.

Our FAM model consists of two parts: the Channel Attention (CA) mechanism, which provides channel-wise weights, and the Pixel Attention (PA) mechanism, which provides pixel-wise weights. The implementation of the Channel Attention mechanism is motivated by [51]. We use an average pooling operation on the input feature $F_{in} \in R^{C \times H' \times W'}$ to obtain the enriched feature $\widetilde{F}_{in} \in R^{C \times 1 \times 1}$; the default setting is $C = 256$. Considering the memory storage, we use $2\times$ convolutional layers instead of fully connected layers in [51] to obtain the channel-wise weight $CA \in R^{C \times 1 \times 1}$. The operation can be expressed as

$$CA = \sigma(Conv_{c2}\delta((Conv_{c1}(\widetilde{F}_{in})))) \tag{3}$$

where $Conv_{c1}$ and $Conv_{c2}$ are the convolution layers with the numbers of kernels $C/8$ and $C$, $\delta$ is the ReLU function and $\sigma$ is the Sigmoid function.

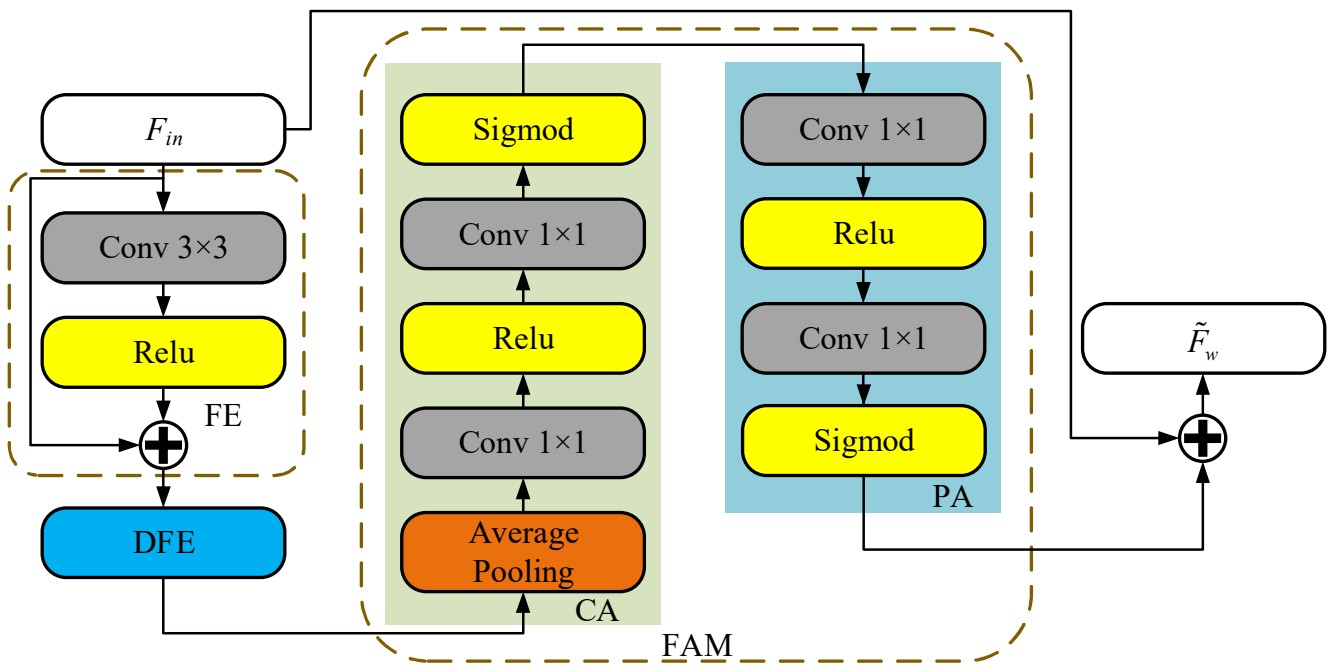

**Figure 2.** The proposed Dynamic Feature Attention module.

Then, we multiply channel-wise the input feature $F_{in}$ and the weights of channel $CA$.

$$\widetilde{F}_c = CA \otimes F_{in} \tag{4}$$

Since the global atmosphere light intensity and medium transmission map are different in each pixel, the distribution of haze is also uneven in the real-world scene. We propose a Pixel Attention mechanism to induce our network to pay more attention to the regions where the haze is thick. Similar to the channel-wise weight, the calculation of pixel-wise weight $PA \in R^{1 \times H' \times W'}$ can be formulated as

$$PA = \sigma(Conv_{p2}\delta((Conv_{p1}(\widetilde{F}_c)))) \tag{5}$$

where $Conv_{p1}$ and $Conv_{p2}$ are the convolution layers with the numbers of kernels C/8 and 1.

The final step is multiplying $\widetilde{F}_c$ and $PA$.

$$\widetilde{F}_w = \widetilde{F}_c \otimes PA \tag{6}$$

Benefitting from the two additional weights provided by FAM, our dehazing network can pay more attention to the important regions.

Most of the latest works [24–27,29,30,47–50], only adopt convolutional layers with fixed kernels to build a network. One work [52] found that convolution layers with spatially invariant kernels could generate over-smoothing artifacts and corrupted image textures, although the dilated convolution can extend the range of the receptive field. It may bring the gridding artifacts in the recovered results. In the dehazing task, the shape of the convolution kernel is also important for feature extraction. Deformable convolution and the shape of the receptive field can be changed by adding an offset parameter to each pixel of the convolution kernel. It has been adopted in some high-level computer vision tasks. As shown in Figure 2, we introduce deformable convolution [31] as the dynamic feature enhancement (DFE) module to expand the receptive field with dynamic kernels and focus the FAM module more on the important regions. In this paper, we use one deformable convolutional layer with kernels $3 \times 3$ and strides 1 as the dynamic feature enhancement module to provide more flexibility for feature extraction. The shape of the features flowing through the DFE module will not change. We place the DFE module before our FAM model.

The experimental results in ablation studies have shown that compared to conventional convolutional layers with spatially invariant kernels, the deformable convolutional layer can significantly improve the performance of the network.

### 3.3. Adaptive Feature Fusion Module

In the encoder–decoder-based network, spatial information from shallow layers will be lost as the layer becomes deeper. Previous works first resample the non-adjacent features to the same shape and fuse them by addition or concatenation. However, the density of information from shallow features and in-depth features is different. Inspired by [29], we designed an Adaptive Feature Fusion (AFF) module based on the adaptive mixup operation [40] to restore the spatial information and fuse the features from non-adjacent levels.

As shown in Figure 3, we describe how to generate the boosted feature from the *nth* AFF in the encoder, and the operation in the decoder can be derived accordingly. The process can be formulated as follows:

$$\widetilde{F}^n = E(F^n, \left\{ \widetilde{F_e}^{n-1}, \dots, \widetilde{F_e}^1 \right\}) \tag{7}$$

where $\widetilde{F}^n$ is the boosted feature from the *nth* AFF module, $F^n$ is the feature from the *nth* downsampling layer of the encoder, and $\left\{ \widetilde{F_e}^{n-1}, \dots, \widetilde{F_e}^1 \right\}$ are the enhanced features generated by fusing $\left\{ \widetilde{F}^{n-1}, \dots, \widetilde{F}^1 \right\}$ and $F^n$. For example, the process of obtaining $\widetilde{F_e}^1$ can be formulated as

$$\widetilde{F_e}^1 = p_e^1(mix(q_e^1(F^n), \widetilde{F}^{n-1})) + F^n \tag{8}$$

where $q_e^1$ and $p_e^1$ are the upsampling operations and the downsampling operations; we stack transposed convolutional/convolutional layers with strides of 2 to implement the upsampling/downsampling operations. The *mix* is the mixup operation similar to Equation (2), which has a learnable balancing parameter. As we have proven that our adaptive mixup connection can improve the dehazing performance with adding ignorable parameters, we also introduce the adaptive mixup connection in the AFF model, which provides additional flexibility in the feature fusion processing without increasing the complexity of the network. By introducing the AFF module, DMADN can leverage all the preceding high-level features to output better results.

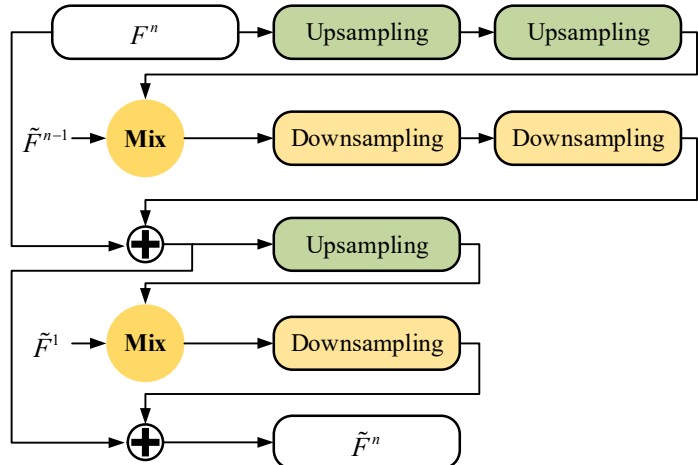

**Figure 3.** Architecture of Adaptive Feature Fusion module.

### 3.4. Loss Function

End-to-end dehazing networks often use L1/L2-based loss function to reconstruct the haze-free image. However, only using L1/L2 loss to approximate the visual performance is

less effective, and the texture details may be lost. We introduce the contrastive learning loss item in training to obtain more natural haze-free results.

Contrastive learning comes from Metric Learning [53]. It aims to build a discriminative representation space to move the output results close to the positive samples while far away from the negative samples. In [54], it was shown that contrastive learning is effective for image translation tasks. Unlike traditional reconstruct loss, contrastive learning could compare the internal consistency of different samples in the representation space. Differently from perceptual learning [55], contrast learning leverages the knowledge from both positive and negative samples. The whole loss function can be formulated as

$$\min\|J - \rho(I)\| + \beta \cdot C(I', I, J) \tag{9}$$

where the first term $\min\|J - \rho(I)\|$ is reconstruction loss; we employ L1 loss since it achieves better performance than L2 loss. $C(G(I'), (G(I), G(J))$ is the contrastive learning regularization item, and $\beta$ is a parameter for balancing the reconstruction loss and contrastive learning item. There are two critical points in the construction of the contrastive learning item: the first is to construct the internal feature space, and the other is to select the suitable pair of positive and negative samples. We introduce pre-trained VGG19 [56] to build the representation space. The contrastive learning item can be formulated as

$$C(I', I, J) = \sum_{i=1}^{n} \omega_i \frac{M(G(J)_i, G(I')_i)}{M(G(I)_i, G(I')_i)} \tag{10}$$

where $I'$, $I$, $J$ are the restored image, hazy input image and ground-truth; we select them as the anchor, positive sample and negative sample, and $\omega_i$ is the weight coefficient. It has been proven in [57] that selecting the knowledge from intermediate features is more efficient in the knowledge distillation task. As shown in Figure 4, we extract the features $G(I')$, $G(I)$ and $G(J)$ from hidden layers of the VGG19 model. We use the Mean Absolute Error function $M(x, y)$ to measure the distance between pairs of samples.

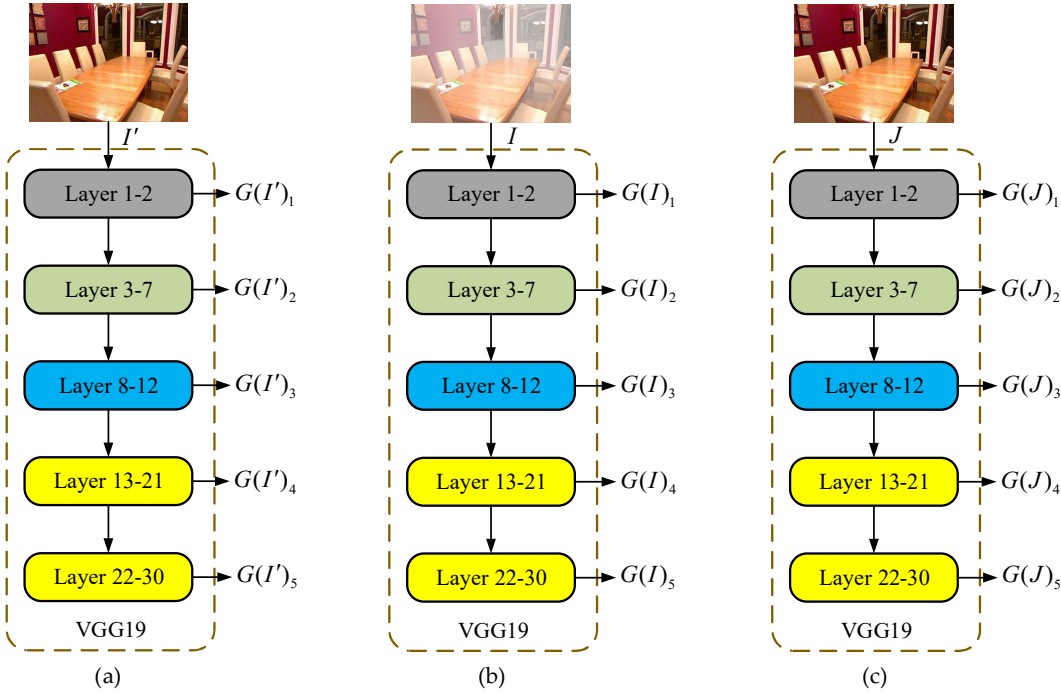

**Figure 4.** We select the intermediate knowledge in VGG19 to construct the internal feature space. We adopt features from the 2nd, 7th, 12th, 21st, and 30th layers of VGG19 model. (**a**) The hidden features of anchors. (**b**) The hidden features of negative samples. (**c**) The hidden features of positive samples.

## 4. Results

In this section, we evaluate DMADN on both dehazing performance and high-level computer vision task results. Our DMADN is trained using Adam optimizer with $\beta_1 = 0.9$ and $\beta_2 = 0.999$ respectively. Because the contrastive learning loss function is challenging to converge, we set the balancing parameter to a low value of 0.1. We also set the initial learning rate to 0.0002, and cosine annealing is used in training. The entire training process contains 200 epochs with a batch size of 12. All experiments are implemented on a PC with one NVIDIA GeForce RTX 3090.

### 4.1. Performance Evaluation

We compare the dehazing performance on both synthetic and real-world datasets. Following previous works, we trained our method on the Indoor Training Set (ITS) and test on the Synthetic Objective Testing Set (SOTS) indoors, which included RESIDE [5]. To verify the dehazing performance of DMADN in the real-world sense, we also adopt O-HAZE [58] and NH-HAZE [59] which are real-world haze datasets. Because dehazing networks are notoriously computationally intensive, it is not sensible to input images with larger resolution in training. On the other hand, the original size of the above datasets is not friendly to the training of dehazing networks. Most recent works [29,46–48] have cropped the training image to square patches. Following [29], we randomly crop $256 \times 256$ patches from training datasets. We also randomly flip the patches horizontally or vertically to augment the training datasets. To compare fairly, we also make the quantitative comparisons on the cropped datasets [29,46–48]. Furthermore, we used SSIM and PSNR for the quantitative assessment.

We evaluate the proposed DMADN against both hand-crafted prior-based methods (DCP [12], NLD [14]) and CNN models (AODNet [24], GridDehazeNet [48], FFA-Net [47], MSBDN [29], AECR-Net [46]). Previous CNN-based dehazing methods only report quantitative assessments of real-world hazy images. To compare fairly, we retrain these methods [24,47,48], using their provided codes on the same datasets as DMADN.

Table 1 summarizes the quantitative assessment of both SOTS, O-HAZE, and NH-HAZE. As we can see, prior-based methods [12,14] do not achieve satisfying performance in quantitative comparisons. The method AOD-Net is less effective because of the overly simple network structure. The previous end-to-end CNN dehazing methods and DMADN obtained better results than others. It can be noticed that DMADN achieves the best performances on both synthetic and real-world datasets. On the STOS dataset, DMADN achieves 37.28 dB PSNR, which gains the second top performance, 0.22 dB PSNR of AECR-Net. On real-world datasets, DMADN gains in the range of 0.31~9.63 dB PSNR and 0.14~6.26 dB PSNR, respectively.

**Table 1.** Quantitative comparisons on both synthetic and real-world datasets.

| Method | SOTS | | O-HAZE | | NH-HAZE | |
|---|---|---|---|---|---|---|
| | PSNR | SSIM | PSNR | SSIM | PSNR | SSIM |
| DCP | 20.76 | 0.8494 | 17.29 | 0.5710 | 14.04 | 0.5003 |
| NLD | 17.27 | 0.7501 | 15.03 | 0.5390 | 13.64 | 0.5551 |
| AOD-Net | 20.23 | 0.8161 | 18.85 | 0.5962 | 15.31 | 0.5796 |
| GridDehazeNet | 32.46 | 0.9794 | 22.94 | 0.6970 | 17.80 | 0.5995 |
| FFA-Net | 35.74 | 0.9846 | 24.20 | 0.7340 | 19.45 | 0.6913 |
| MSBDN | 33.79 | 0.9840 | 24.35 | 0.7485 | 19.23 | 0.7056 |
| AECR-Net | 37.06 | 0.9898 | 24.24 | 0.7480 | 19.76 | 0.7172 |
| Ours | 37.28 | 0.9913 | 24.66 | 0.7502 | 19.90 | 0.7175 |

We also show the qualitative comparison of both synthetic and real-world hazy images. Figure 5 shows that the restored images of DCP and NLD have a color distortion problem, and AOD-Net cannot remove all haze in synthetic images. The recent CNN-based dehazing methods and DMADN can all effectively remove haze and preserve textures well.

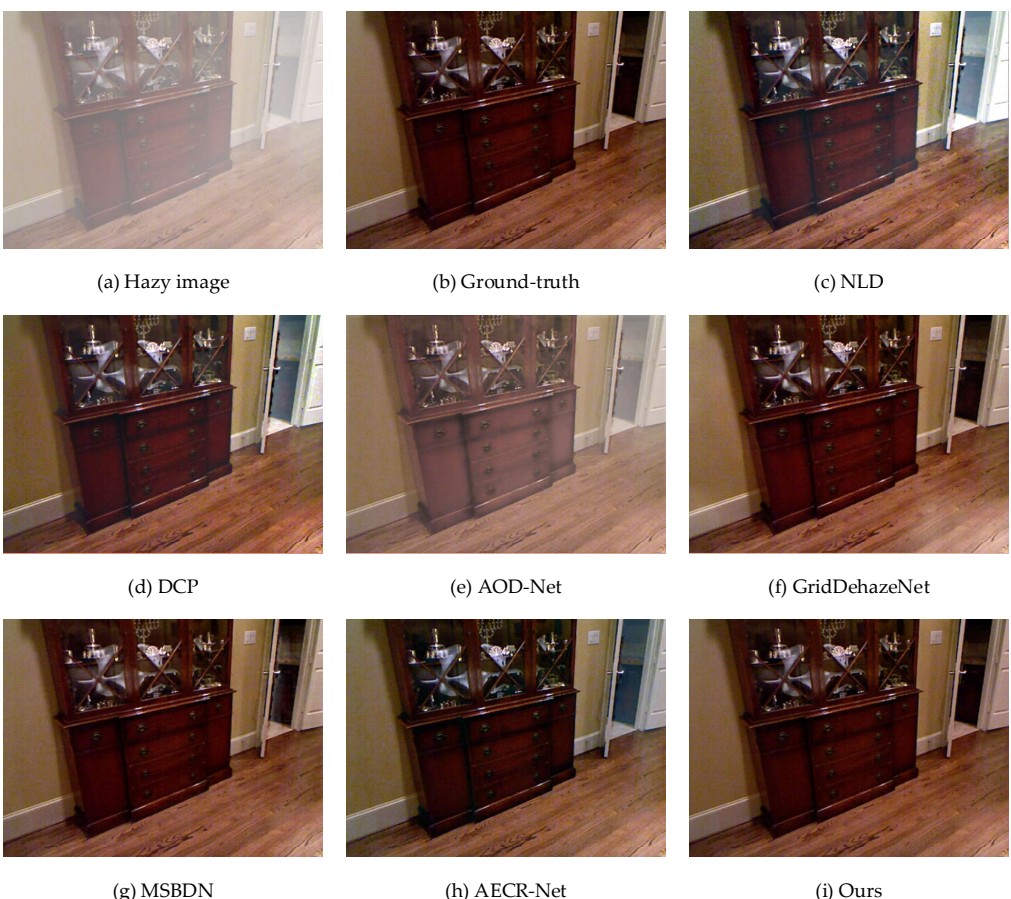

**Figure 5.** Visual quality comparison of SOTS dataset.

We present the visual quality on O-HAZE and NH-HAZE in Figures 6 and 7. Compared to synthetic images, real-world hazy images are more difficult to recover, since the natural haze may lead to severe degradation of information. As we can see, the images generated by DCP and NLD suffer from severe color distortion. The images restored by AOD-Net and GridDehazeNet still have visible haze. The restored haze-free images by DMADN are sharper and brighter.

To demonstrate the applicability of our DMADN, we further evaluate our method on a natural hazy image. Figure 8 shows a natural hazy image and the recovered results from state-of-the-art dehazing methods [12,24,29,48]. As we mentioned, the dehazed image generated by DCP suffers from significant artifacts in the white areas, and AOD-Net generates the results with significant color distortions. The restored result by our DMADN is brighter and clearer.

*4.2. Perceptual Quality Comparsion for High-Level Computer Vision Task*

It has been noticed that the results of high-level computer vision tasks (such as object detection and tracking) may be poor when the input images are degraded. Furthermore, the dehazing methods could be used as a pre-processing module to improve the robustness of high-level computer systems. Such a "task-driven" evaluation can be used as an indirect indicator of dehazing performance. In this paper, we conduct an object detection performance study to indirectly evaluate the dehazing performance.

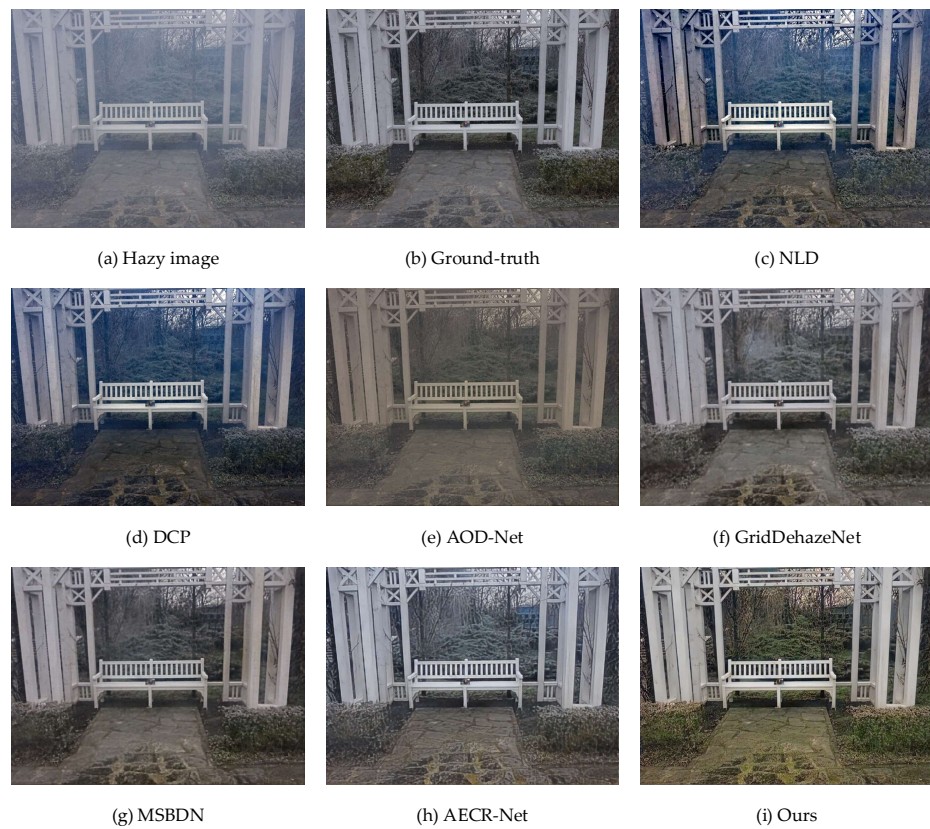

**Figure 6.** Visual quality comparison of O-HAZE dataset.

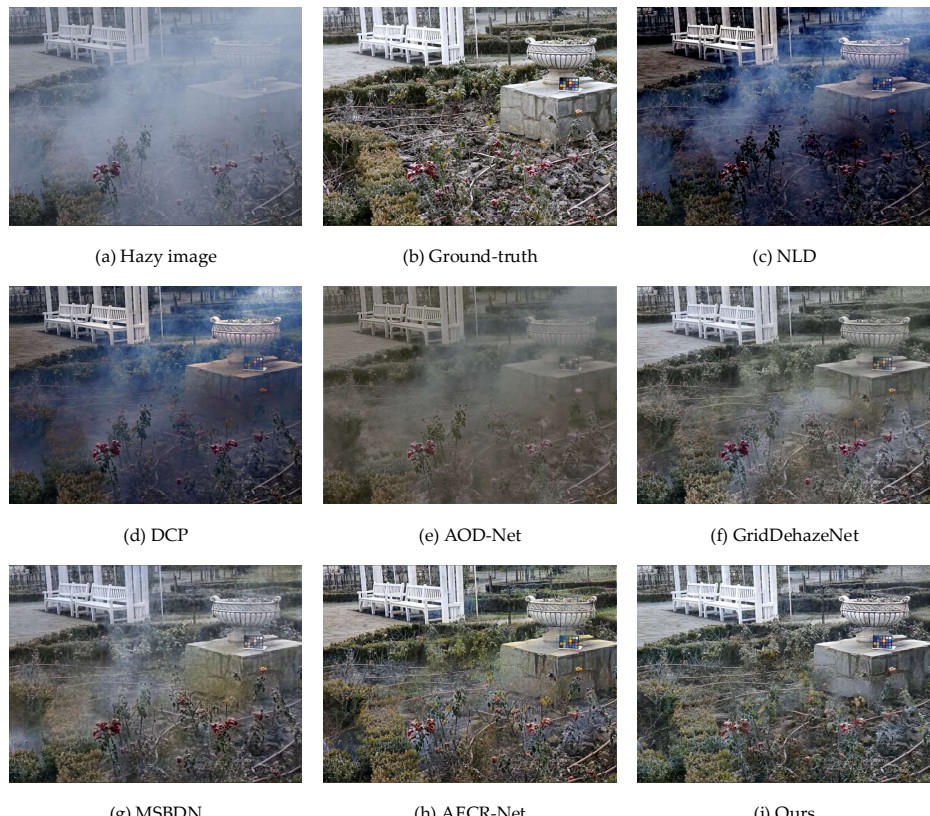

**Figure 7.** Visual quality comparison of NH-HAZE dataset.

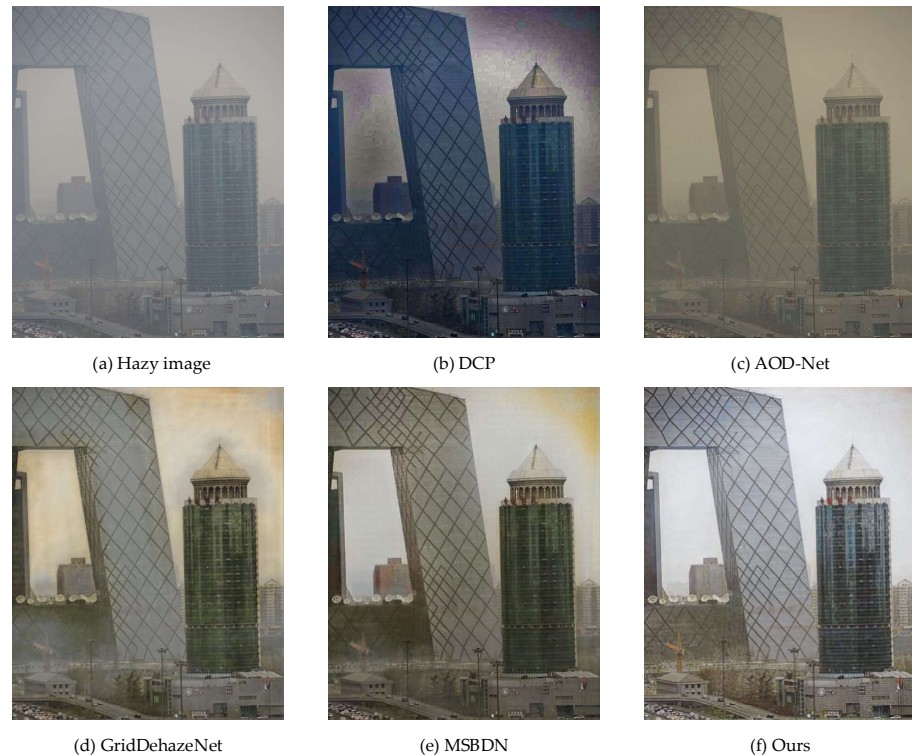

**Figure 8.** Visual quality comparison of natural hazy image.

Though it is meaningful to study the application of the objection detection task in hazy conditions, it has not attracted enough attention. There are also few publicly available datasets for object detection in haze conditions. Although RTTS provides a hazy dataset with labeling, it lacks corresponding clear images for training. As shown in Figure 9, we built upon the VOC dataset [60] with a VOC_hazy_test dataset according to [5] to conduct fair evaluation. Following RTTS, we selected five classes of objects (person, car, bicycle, motorcycle, bus) to build our VOC_haze_test. In the perceptual quality evaluation, we first trained a YOLOv5s model on clear VOC2007 and VOC2012 datasets, then we used the trained model to evaluate the recovered haze-free images. In the "task-driven" evaluation, we compare the proposed DMADN with DCP [12], PFF-Net [47] and MSBDN [29].

The mAP results are demonstrated in Table 2, and our DMADN obtains the best performance. An example of qualitative performance is shown in Figure 10. Benefitting from DMADN, the lost objects have been recovered. It means that our DMADN can not only recover images with the best structural similarity, but also is helpful in improving the robustness of the detection algorithm.

**Table 2.** Detection results on recovered images.

|  | **Person** | **Car** | **Bus** | **Bicycle** | **Motorbike** | **All** |
|---|---|---|---|---|---|---|
| Hazy | 81.7 | 86.8 | 79.4 | 82.0 | 73.0 | 80.6 |
| DCP | 84.5 | 86.0 | 84.6 | 85.5 | 82.4 | 84.6 |
| PFF-Net | 79.5 | 85.6 | 80.9 | 80.9 | 77.9 | 81.0 |
| MSBDN | 85.6 | 89.9 | 85.5 | 87.8 | 84.5 | 86.7 |
| Ours | 87.4 | 91.0 | 87.3 | 89.0 | 86.6 | 88.4 |

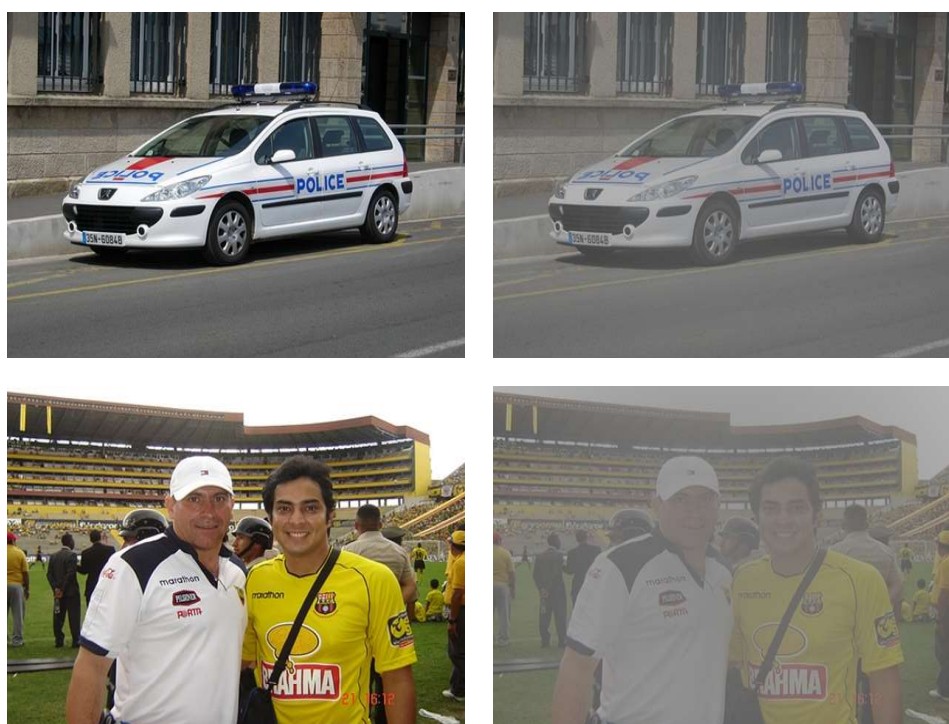

**Figure 9.** Example of VOC_hazy_test dataset.

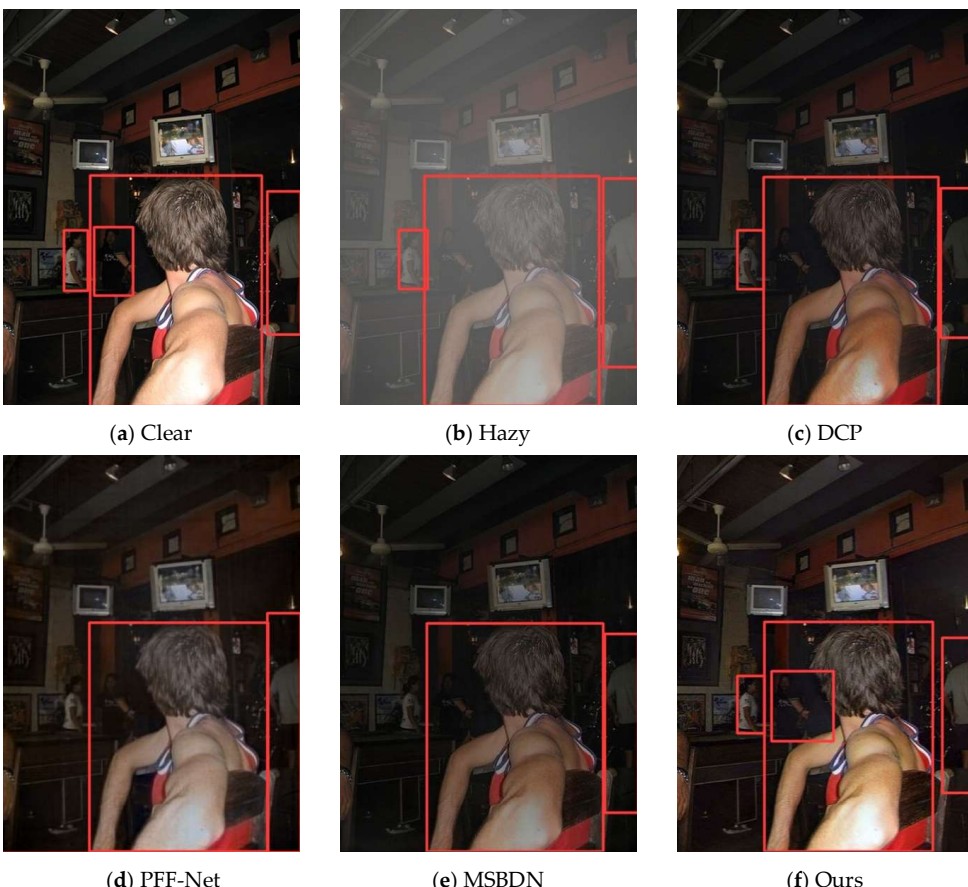

(**a**) Clear　　　　　　　(**b**) Hazy　　　　　　　(**c**) DCP

(**d**) PFF-Net　　　　　　(**e**) MSBDN　　　　　　(**f**) Ours

**Figure 10.** Detection results on the recovered synthetic hazy images.

*4.3. Ablation Studies*

To demonstrate the effect of different models in DMADN, we conduct ablation experiments in this part.

We first built a base subnet as the baseline, consisting of only downsampling layers, $6\times$ FE modules, and upsampling layers. Then, we added different components into the basic network: (1) **base + mix**: We added the learnable adaptive mixup skip connection into the base subnet. (2) **base + mix + FAM**: Adaptive mixup skip connection and Feature Attention Mechanism were added into the base subnet. A conventional convolutional layer replaces the DFE module. (3) **base + mix + DFA**: The network includes adaptive mixup skip connections and a complete DFA module. (4) **DMADN**.

In the ablation studies, we use ITS for training and SOTS indoor for testing; a comparison of the above subnets is shown in Table 3. Compared with the baseline, base + mix achieves 0.88 dB PSNR gains at adding ignorable parameters.

**Table 3.** Ablation study on DMADN.

| Subnet | PSNR | SSIM | Param |
|---|---|---|---|
| base | 28.92 | 0.9494 | 1.34M |
| base + mix | 29.80 | 0.9562 | 1.34M |
| base + mix + FAM | 33.49 | 0.9797 | 1.96M |
| base + mix + DFA | 35.62 | 0.9854 | 2.00M |
| DMADN | 37.28 | 0.9913 | 5.34M |

The results in Table 3 can demonstrate the importance of the FAM module. We have proven that the introduced DFE module dramatically enhances the dehazing performance by improving 2.13 dB PSNR from base + mix + FAM to base + mix + DFA. Moreover, compared to base + mix + DFA, our DMADN achieves performance gains of 1.66 dB PSNR, demonstrating that the AFF module can restore the missing spatial information efficiently.

To demonstrate the effect of the contrastive learning mechanism, we also trained the DMADN on ITS by only employing the L1 loss. As presented in Table 4, the contrastive learning mechanism improves the performance gains of 1.88 dB PSNR without increasing inference time in the testing phase.

**Table 4.** Ablation study on contrastive learning.

| Model | Contrastive Learning | PSNR | SSIM |
|---|---|---|---|
| DMADN | $\checkmark$ | 37.28 | 0.9913 |
| DMADN | $\times$ | 35.40 | 0.9831 |

## 5. Discussion and Conclusions

This paper proposes a novel single-image dehazing network named DMADN. Extensive experiments have demonstrated that DMADN achieves state-of-the-art dehazing performance with competitive complexity. The perceptual quality evaluation for object detection has proven the application value of our method.

Our DMADN is designed under the encoder–decoder framework; it consists of two key components, the Dynamic Feature Attention (DFA) module, and the Adaptive Feature Fusion (AFF) module. We took advantage of the characteristic of haziness in designing networks. Considering that the distribution of haze is always uneven in a degenerated image and the value in each channel is different [12], we designed the Dynamic Feature Attention module, which provides pixel-wise weights and channel-wise weights for input features. Benefiting from the DFA, our dehazing network can focus on the more important regions in the feature maps. In the ablation studies, we demonstrated that DFA can effectively improve the dehazing performance. We believe that this strategy of adding extra weight to features could be extended for future research on dehazing networks. We notice

that previous dehazing methods only adopted convolutional layers with fixed kernels to build networks, which cannot extract spatial structure information in the input images. One work [52] found that convolution layers with spatially invariant kernels could generate over-smoothing artifacts and corrupted image textures. To extract the information more efficiently, we introduce deformable convolution [31] as the dynamic feature enhancement (DFE) in our DFA module. Our design idea is that by using the deformable kernel of deformable convolutional, we can move the receptive field to the more important parts (such as an edge part of an object). The ablation study results show that compared with the conventional convolutional layer, the deformable convolution achieves 2.13 dB PSNR on SOTS. Due to the significant improvement over the deformable convolution, it is reasonable to assume that it could also be applied in future research on image restoration. It is an easy problem to achieve balance between the number of parameters and the network's performance. Unfortunately, in order to obtain a natural result, it is important to increase the depth of the network to better extract the information of semantic features. To deal with this problem, we did not simply adopt multiple independent CNN models. Following works [29,51], we define only one DFA module in the feature restoration module, and we let the input features pass through this defined DFA module several times. This approach of using shared parameters in the network slightly decreases performance compared to using separate parameter modules. However, its semantic extraction power is significantly enhanced compared to using the corresponding convolutional layer only once. We believe this is an effective parameter-saving approach applicable to other works.

Most end-to-end dehazing networks are typically designed under an encoder–decoder framework. Moreover, the spatial information from the higher layers is degenerating as the convolutional layers are stacked. To resume the missing information, previous works first resize the non-adjacent features to the same dimension and fuse them via skip connection. However, simply using skip connection or conventional convolutional layers to accomplish feature fusion is inefficient since the density of information from shallow and in-depth features is different. In this paper, we design a non-adjacent feature fusion module named Adaptive Feature Fusion (AFF) to remedy the missing spatial information. In addition, we have proven that the adaptive mixup operation can improve the dehazing performance by adding ignorable parameters, and introduced the adaptive mixup operation in our AFF module. Our AFF model has a high degree of versatility and can be easily applied to other dehazing networks with similar structures. The experimental results in ablation studies have proven that our proposed AFF module is practical for improving the dehazing performance.

Different from object detection CNNs, dehazing networks are notoriously computationally intensive. Most previous works concentrated on increasing the scale of the model to improve the dehazing performance, making it challenging to apply dehazing methods in edge devices. In this paper, we introduce contrastive learning to leverage the knowledge from both positive and negative samples. Furthermore, we have proven that the contrastive learning strategy could effectively improve the quality of output while adding no additional complexity and inference time in the testing phase. This scheme of improving the performance without increasing the network's complexity is friendly for the application of edge devices.

Compared to conventional convolution, the deformable convolution has an additional offset parameter at each pixel of the kernels. This characteristic of the deformable convolution may create a confusion about whether, in the testing phase, the sizes of input images have an impact on the dehazing performance. In general, the end-to-end dehazing network built with only conventional convolutional layer usually achieves the best performance on the testing image with the same size as training dataset. To verify this view, we use the trained subnet in the Section 4.3 to evaluate on cropped testing datasets with different sizes. In the experiment we find that the subnet base + mix + FAM achieves the best performance on the testing dataset with the size $256 \times 256$. Compared to the results on images with size $256 \times 256$, the results on the hazy image with smaller or larger size are decreasing.

This result can prove the above point. However, the result of subnet base + mix + DFA is different from the base + mix + FAM. As the size of the input image increases, the result of the quantitative comparison will become higher. Moreover, the decline on smaller size is lesser than base + mix + FAM. The only difference between above two subnets is that the deformable convolutional layer in base + mix + DFA has been replaced by a conventional convolutional layer in the base + mix + FAM. So, we can theorize that the deformable convolution causes the condition. We interpret the above experimental results as the reason that the deformable convolution introduces additional parameters; its performance on larger-size images may be better than its application on lesser-size images. Furthermore, these additional parameters may give it some robustness to input size variations. This characteristic of deformable convolution brings two advantages to our dehazing network. One of them is that we can directly input large images into the trained model without worrying about significant performance degradation. On the other hand, it suppresses the deterioration of the model's dehazing performance on small-sized input images.

To make our dehazing network more suitable for practical applications, we have added a compatible function for the different sizes of input images. Since there are two models (adaptive mixup operation and Adaptive Feature Fusion Module) used to fuse the non-adjacent features in our network, when the input image size is not $2^n$, it may produce an error in the forward process of the network. To deal with this problem, we add interpolation operations in DMADN. Unlike the interpolated operation, which directly uses the input images, the effect of the interpolation operation in the network can be ignored. In fact, due to deconvolution and skip operation, most recent end-to-end dehazing networks have the problem of not being compatible with the image with the size is not $2^n$. Through the compatible function, anyone who wants to use our trained model can input the images directly into our model without cropping.

It has been noticed that the results of high-level computer vision tasks may deteriorate the haze. Dehazing methods could be used as the pre-processing for high-level computer systems. Although the "task-driven" evaluation has tremendous implications for practical applications, it has received little attention. To illustrate the superiority of our model, we not only evaluate DMADN against state-of-the-art dehazing performance and result of the object detection task. There is a lack of detection datasets containing both clear and hazy images. To evaluate the perceptual quality, we used [5] to build the VOC_haze_test. We evaluate the object detection performance of our DMADN and other dehazing methods. Our models achieved the highest scores in all object detection experiments, meaning that the DMADN has a high application value. We believe that this reference-free evaluation approach could not just be adopted in terms of dehazing, but also could be used in the evaluation of other image recovery tasks (such as denoising and deblurring). On the other hand, we hope that the studies for high-level computer vision tasks in degraded environments can receive more attention.

**Author Contributions:** Conceptualization, D.Z.; methodology, D.Z. and B.M.; funding acquisition, D.Z. and B.M.; validation, D.Z. and X.Z.; formal analysis, D.Z. and J.Z.; investigation, J.Z., Y.T. and C.Z.; data curation, D.Z., H.Z. and Y.T.; writing—original draft preparation, D.Z.; writing—review and editing, D.Z.; visualization, D.Z., X.Z. and H.Z. All authors have read and agreed to the published version of the manuscript.

**Funding:** This research received no external funding.

**Data Availability Statement:** The data presented in this study are available on reasonable request from the corresponding authors.

**Conflicts of Interest:** The authors declare no conflict of interest.

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
