# Peer review of "Dynamic Multi-Attention Dehazing Network with Adaptive Feature Fusion"

_electronics, doi:10.3390/electronics12030529_

Round 1

Reviewer 1 Report

In MDPI Electronics

"Dynamic Multi-Attention Dehazing Network with Adaptive Feature Fusion" by

Donghui Zhao, Bo Mo, Xiang Zhu, Jie Zhao, Heng Zhang, Yimeng Tao, Chunbo Zhao

A simplified Unet architecture with special

fusion bridges between encoder and decoder is defined for image dehazing application. It is called Dynamic Multi-Attention Dehazing Network (DMADN).

Authors claim to propose SOTA image dehazing solution. However, the reviewer observes serious missing parts and drawbacks of the submitted paper.

1. DFE acronym is mentioned in Figure 2 as the part of Dynamic Feature Attention module, but in the main text beside saying that it denotes Dynamic Feature Enhancement (DFE) module, being related to a deformable convolution, we get no more details. However, it seems that this is crucial unit of DMADN, as being repeated 6 times, it extends the number of parameters by 6/16 of the number of pixels in the input image. It makes the status of the proposal "it is resolution dependent" - rather undesirable property for image dehazing algorithm.

2. FAM acronym is explained just after decrypting of DFE acronym as the Feature Attention Mechanism (FAM) module. In line 193 we learn only that FAM is placed in the architecture just after DFE module, but it is not shown in Figure 2 at all. However, this module is even more crucial than DFE module since in comparative study given by authors it plays important role. In the DMADN architecture evolution it is mentioned at least four times.

3. The conclusion on chaotic final edition of the paper written by seven authors, can be supported by few detailed observations:

(a) In Figure 2 we have two skip connections ending in adders of parts FE and PA, respectively, but with no their beginning point at all. Such residual connections are really physically not feasible.

(b) The Mix unit, used by authors in many fusion points of the DMADN architecture, in fact being the simple convex combination of the input feature tensors with learned weights, is referenced by them to a 2018 paper describing a procedure for training data augmentation by convex combination with weights which are not learned but produced randomly from Beta distribution. It would lead a potential reader to quite wrong intuitions.

(c) In the paper, the original images and its hazed versions imply its synthetic form. As this makes the sense for training, for testing we want to see also images of natural haze without its original images and the dehazed results produced by DMADN architecture. Otherwise, someone can claim that DMADN learns only synthetic haze.

Due to the above paper drawbacks, the paper in its current form cannot be evaluated with respect to its important aspects. Therefore, I suggest to make a new, more accurate edition of the paper and then its resubmission.

Reviewer 2 Report

Dear authors

There are some suggestions to improve the work further:

1. The introduction is a bit short. Also, it needs improvement. The gaps in the prior state of research should be shown clearly, which is not the case right now. As an example, authors state “We notice that most end-to-end dehazing networks are typically designed under an encoder-decoder framework, which is inherently limited to one problem”. How did you notice this. Where are the examples and references to substantiate this claim. This is just one example. You have to further build on the introduction to show the gaps and build the significance of this study.

2. Related works section should be enhanced by including more related works. The examples and the references given in this section are inadequate.

3. Further, the study has also 39 references, which is not appropriate for the area of research. Authors need to include more works, giving preference to the works published in the last 3 to 5 years.

4. The study's implications should be discussed in greater detail in the discussion and conclusion. This would also allow the reader to understand the significance of this study.

Round 2

Reviewer 1 Report

Authors considered my comments in an adequate way, explaining perhaps all doubts raised in my original opinion.

For me now only one concern remains: dependence of parameter number P in the final model on the input image resolution.

The concern follows from the fact that using DFE on the second level of downsampling in the encoder you get 9x2x(P/(4x4))>P extra parameters for deformable convolution which is position sensitive.

Apparently this parameter overhead depends on the input image resolution.

Therefore few question can be addressed:

- What is the nominal resolution of the input image if the dehazing model is given to users?

- What are results if a user of dehazing application adjusts their image resolution to the nominal one?

- What are results if on the other hand your application interpolates all local deformations to fit the current resolution and then no action from the user is required?

Be so kind to add the answers to the above questions in the final version of your paper.
